# Genomic epidemiology of SARS-CoV-2 in Russia reveals recurring cross-border transmission throughout 2020

Alina Matsvay[1◉], Galya V. Klink[2◉], Ksenia R. Safina[3], Elena Nabieva[3], Sofya K. Garushyants[2], Dmitry Biba[3], Georgii A. Bazykin[2,3]*, Ivan M. Mikhaylov[1], Anna V. Say[1], Anastasiya I. Zakamornaya[1], Anastasiya O. Khakhina[1], Tatiana S. Lisitsa[1], Andrey A. Ayginin[1], Ivan S. Abramov[1], Sergey A. Bogdan[4], Kseniya B. Kolbutova[4], Daria U. Oleynikova[4], Tatiana F. Avdeenko[4], German A. Shipulin[1], Sergey M. Yudin[1], Veronika I. Skvortsova[5]

1 Federal State Budgetary Institution "Centre for Strategic Planning and Management of Biomedical Health Risks" of the Federal Medical Biological Agency, Moscow, Russia, 2 A.A. Kharkevich Institute for Information Transmission Problems of the Russian Academy of Sciences, Moscow, Russia, 3 Skolkovo Institute of Science and Technology (Skoltech), Moscow, Russia, 4 Chief Federal State Budgetary Healthcare Institution "Centre of Hygiene and Epidemiology" of the Federal Medical Biological Agency, Moscow, Russia, 5 Federal Medical Biological Agency, Moscow, Russia

◉ These authors contributed equally to this work.
* g.bazykin@skoltech.ru

**Data Availability Statement:** All relevant data are within the paper and its Supporting Information files.

## Abstract

In 2020, SARS-CoV-2 has spread rapidly across the globe, with most nations failing to prevent or substantially delay its introduction. While many countries have imposed some limitations on trans-border passenger traffic, the effect of these measures on the global spread of COVID-19 strains remains unclear. Here, we report an analysis of 3206 whole-genome sequences of SARS-CoV-2 samples from 78 regions of Russia covering the period before the spread of variants of concern (between March and November 2020). We describe recurring imports of multiple COVID-19 strains into Russia throughout this period, giving rise to 457 uniquely Russian transmission lineages, as well as repeated cross-border transmissions of local circulating variants out of Russia. While the phylogenetically inferred rate of cross-border transmissions was somewhat reduced during the period of the most stringent border closure, it still remained high, with multiple inferred imports that each led to detectable spread within the country. These results indicate that partial border closure has had little effect on trans-border transmission of variants, which helps explain the rapid global spread of newly arising SARS-CoV-2 variants throughout the pandemic.

## Introduction

The earliest known case of COVID-19 was admitted to a hospital in Wuhan, China, on December 16, 2019 [1], and the first case outside China was reported on January 13, 2020 [2]. In Russia, the first case was reported on March 1, 2020 in a patient returning from Italy [3]. By

**Funding:** This study was partially supported by RFBR project 20-04-60556. The funders had no role in study design, data collection and analysis, decision to publish, or preparation of the manuscript.

**Competing interests:** The authors have declared that no competing interests exist.

the time Russia closed its borders on March 30, 2020, 1836 cases were reported in Russia [4] and 888,460 globally [5]. SARS-CoV-2 has spread rapidly throughout Russia, with evidence for tens of introductions by summer 2020 [6, 7]. COVID-19 cases in Russia have spiked in May and then again near the end of 2020, with a total of 3,612,800 documented cases as of January 19, 2021 [5] (S1 Fig).

After border closure at the end of March 2020, international air traffic from and to Russia was reduced from four-five million passengers per month in January-February to ~30,000 in April [8] (Table A in S1 Text). The border was gradually reopened starting from early August [9]; accordingly, numbers of travellers increased, reaching ~1.5 million per month in September and October 2020 [8] (Table A in S1 Text).

Here, we detail the dynamics of COVID-19 variants in Russia between March-November 2020. We focus on the period prior to the spread of variants of concern; over this period, variants have not substantially differed in transmissivity, simplifying analysis. We show that the intensity of cross-border transmission remained high throughout this period, with multiple transfers of variants in both directions.

## Results

### Diversity of SARS-CoV-2 in Russia

To study the diversity of SARS-CoV-2 in Russia, we sequenced complete viral genomes from 1636 samples obtained between March 2 and November 25, 2020. We combined this data with the 1570 genomes from Russia for the period of March 11—November 28 that we downloaded from GISAID on January 5th, 2021. We also included one sample obtained in Japan on February 25 from a passenger from the Diamond Princess cruise ship who later returned to Russia [10]; and 15 samples from Kazakhstan obtained from Russian citizens. The resulting dataset included 3206 sequences from 78 out of the 85 regions of the Russian Federation (S2 and S3 Figs). The merged datasets did not overlap.

The Pango lineage composition of the Russian sequences was distinct from the world-averaged composition of sequences in GISAID at the same times (Fig 1). The ancestral B.1.1 lineage has prevailed in Russia throughout the study period, while globally, it has been largely displaced by other descendants of B.1 (B.1.* on Fig 1) as well as lineages descendant to B.1.1 (B.1.1.* on Fig 1). Prior to the onset of the variants of concern, there has been little difference in transmissivity between the spreading lineages. Therefore, differences in lineage diversity between countries and their dynamics has been mainly due to genetic drift rather than selection [11, 12]. Indeed, there is no evidence of fitness differences between lineages circulating in Russia [13, 14].

To study the origin of SARS-CoV-2 in Russia in more detail, we reconstructed a maximum likelihood phylogeny of SARS-CoV-2 that comprised 3,206 Russian and 120 226 non-Russian sequences. For this, we obtained from GISAID all available SARS-CoV-2 sequences, aligned and filtered them, and joined them with our dataset (see Methods for details). To facilitate analysis, we then collapsed closely related non-Russian sequences. The final dataset was divided into five major SARS-CoV2 PANGOLIN lineages circulating at that time, and separate phylogenetic trees were built and analyzed for each subset (see Methods).

Following previous work [6, 15], we then grouped Russian sequences into Russian transmission lineages on the basis of their positions in the phylogenetic tree. Each Russian transmission lineage was defined as a monophyletic group (clade) carrying more than one sequence all of which are characterized by common mutations and are Russian, indicating within-Russian transmission. A single introduction of a viral variant into Russia can result in one or more Russian transmission lineages. The remaining samples were as much or more related to non-

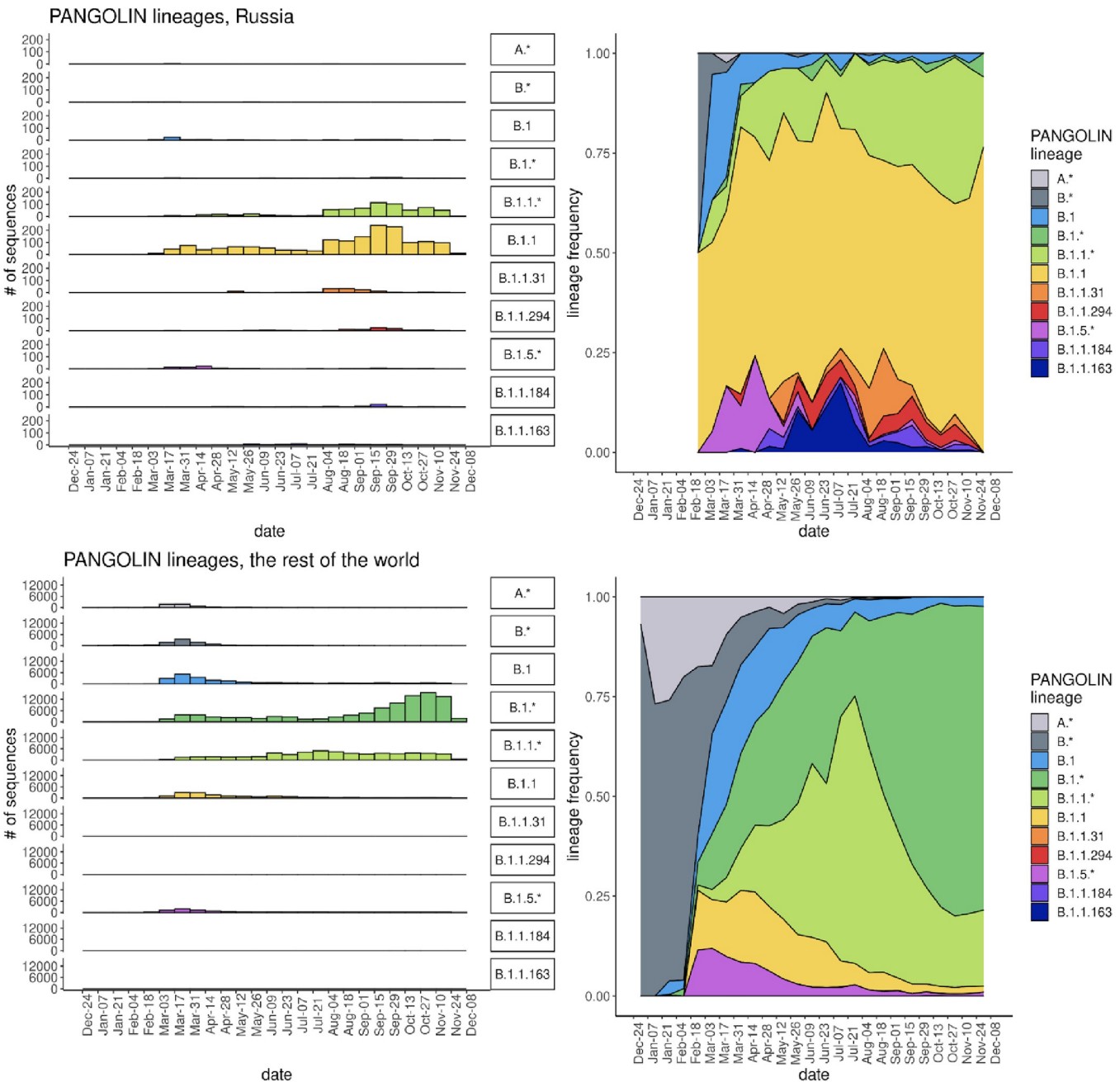

**Fig 1.** Dynamics of Pango lineages frequencies in Russia (top row) and in the rest of the world (bottom row) before November 2020. Sequences are ordered by sampling date, split into 14 day bins. Color legend as in Fig 1.

Russian sequences as to the Russian ones, and were therefore not grouped into Russian transmission lineages. Such sequences were classified as "singletons" if they carried their own mutations, or as "stem clusters" if they did not [6] (Fig 2A and 2B).

We identified 457 uniquely Russian transmission lineages, together encompassing 2089 (65%) of the sequences. The remaining sequences represented singletons (11%) or stem clusters (24%). Many of the Pango lineages, notably the B.1.1 lineage, crossed the Russia's border repeatedly (Fig 3). The earliest sampling dates for Russian transmission lineages fall

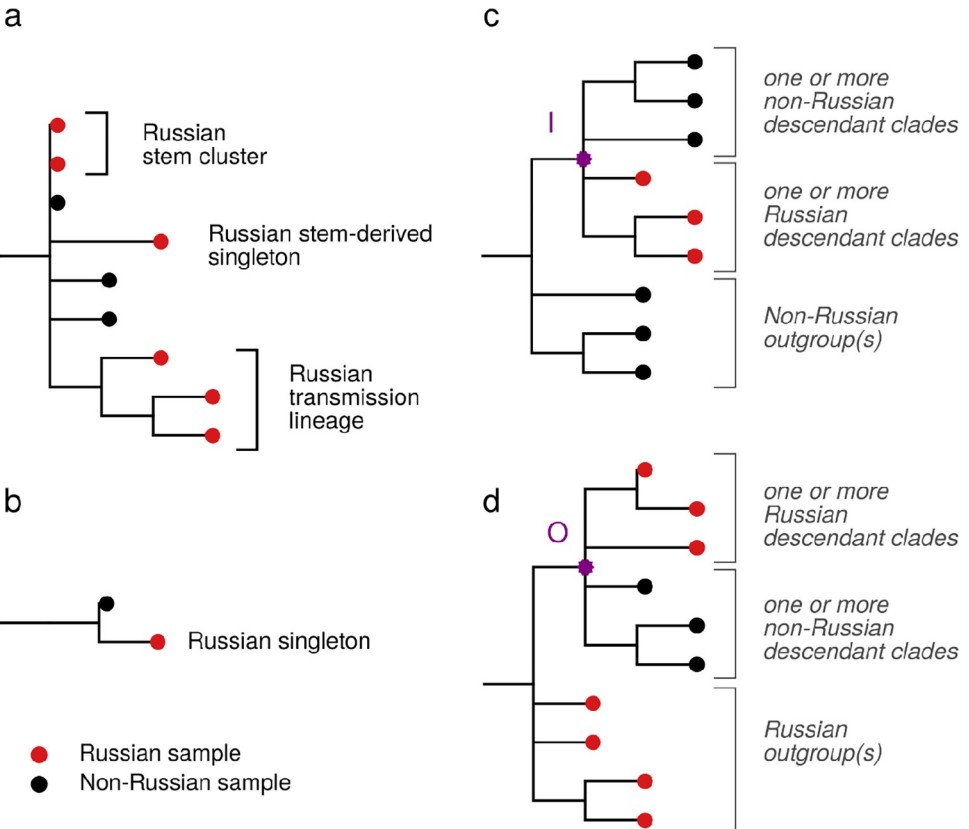

**Fig 2.** Phylogenetic categorization of sequences into Russian transmission lineages, stem clusters, stem-derived singletons (a) and singletons (b), and inference of IBTs (c) and OBTs (d). I, node ancestral to an IBT event; O, node ancestral to an OBT event.

throughout the study period (Fig 3). The number of Russian transmission lineages continued to increase with the number of obtained sequences with little evidence for saturation (S4 Fig), suggesting that some lineages remained undetected and/or that lineages continued to be imported.

Overall, Russian transmission lineages were rather well mixed within Russia. 216 of the 454 lineages with known sampling locations (47,6%) were each observed in two or more regions, suggesting intensive transport of individual lineages within Russia (S5 Fig). The remaining 238 lineages were limited in their spread to just one region.

## Cross-border transmission of SARS-CoV-2 strains

We asked how SARS-CoV-2 lineages were imported into Russia (inbound transmissions, IBT) and transmitted out of it (outbound transmissions, OBT). Our parsimony-based definitions of IBT and OBT are based on the intuition that a Russian clade surrounded by non-Russian sequences might arise as a result of cross-border transmission to Russia, and vice versa (Fig 2C and 2D, see Methods). To obtain conservative estimates for numbers of IBTs and OBTs, we defined all sister Russian lineages descendant from a non-Russian ancestor as results of a single IBT, and all sister non-Russian lineages descendant from a Russian ancestor, as results of a single OBT (see Methods, Fig 2C and 2D). Under these definitions, we observe a total of 82 IBTs, and 43 OBTs. These numbers are the lower boundaries; in particular, multiple Russian transmission lineages derived from a single stem node are here considered as a result of a single

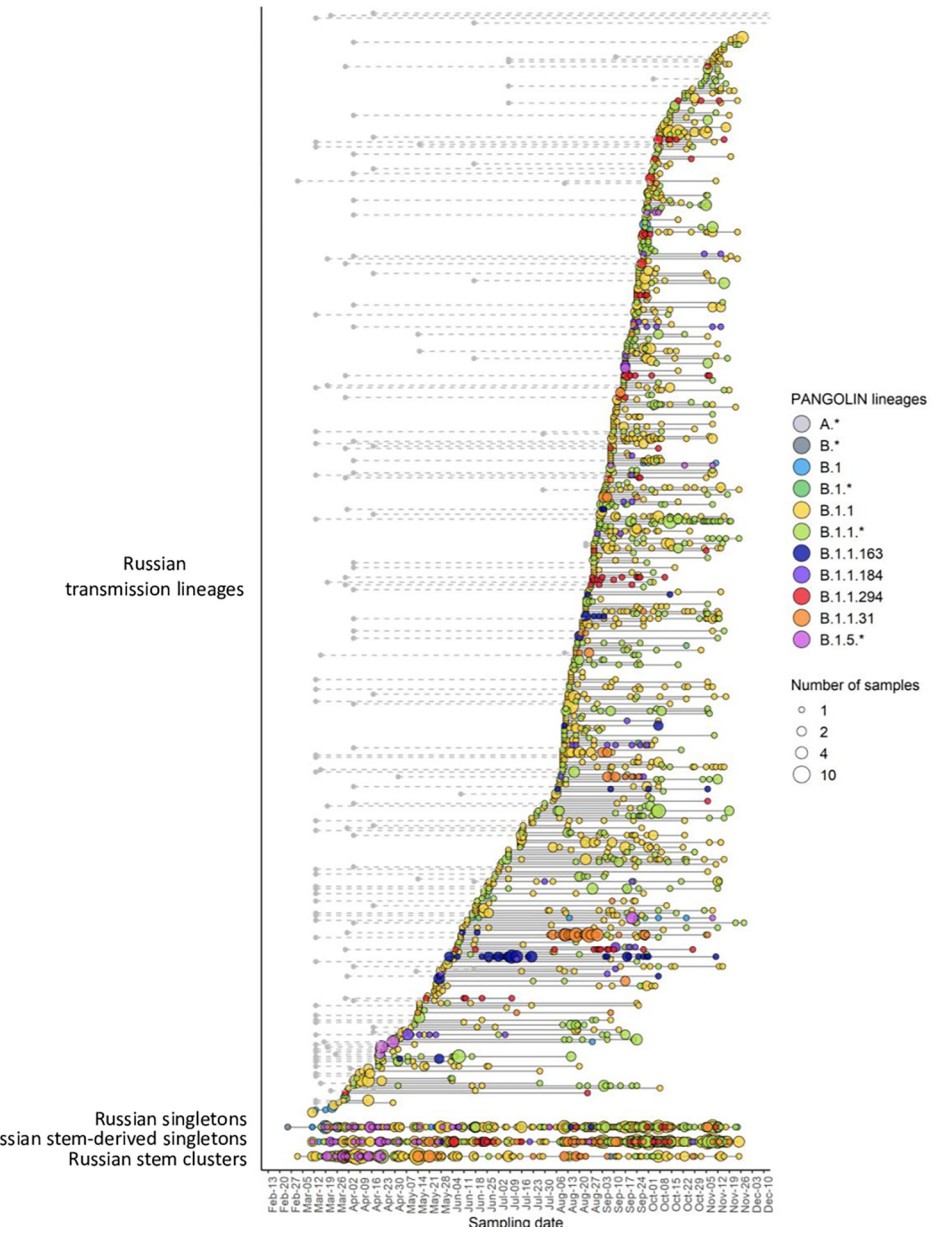

**Fig 3. Russian transmission lineages.** In the top part of the figure, each horizontal line represents a Russian transmission lineage, ordered by the date of the earliest sample. For Russian transmission lineages descended from a Russian stem cluster, the small gray circle in the left identifies the earliest Russian sequence date for the corresponding stem cluster. The three bottom lines represent Russian singletons, Russian stem-derived singletons, and Russian stem clusters (see S2 Fig for terminology for lineage notation). Circles represent samples taken on a particular date, with circle size representing the number of samples. Circle color indicates the PANGOLIN designation of the corresponding sample. Gray lines without colored terminal nodes are transmission lineages without dated sequences.

IBT. An alternative definition of IBTs and OBTs obtained using Treetime [16] (S6 Fig; see Methods) yielded larger numbers of inferred IBTs (418) and OBTs (118); nevertheless, these lists included 94% (118 of 125) of events detected using our definition, and all of the seven events absent in the Treetime-based list could be also inferred from the Treetime output with a lower probability threshold (0.75 instead of 0.8; see Methods), indicating robustness of inferred IBTs and OBTs.

Overall, we find that transmissions of SARS-CoV-2 variants into Russia which started in spring 2020 continued throughout summer and fall 2020, despite interventions aimed at curbing transborder travel. Similarly, we observe likely transmissions out of Russia starting in spring, and continuing throughout the summer and fall (Fig 4).

To better understand the effect of border closure on the rate of inferred IBTs (OBTs), we dated these events using two approaches: by the maximum likelihood estimation of the last common ancestor of Russian (non-Russian) sequences descendant from these events (see Methods), or by the earliest sampling date of these sequences (Fig 4). Using each approach, we subdivided cross-border transmissions (IBTs and OBTs) into two groups: those that dated to the period from the first transmission to border closure (between 23.01.2020 and 29.03.2020 for ML, and between 11.03.2020 and 29.03.2020 for the earliest sampling dates), and those that dated to the period of the most intensive travel restrictions (30.03.2020 to 01.08.2020). We then asked if the rate of inferred cross-border transmissions differed between these two intervals.

Under both dating methods, the estimated rate of cross-border transmission was substantially reduced by border closure. An IBT or an OBT was observed almost three times more often prior to 30.03.2020 (once every 1.6 days) than between 30.03.2020–01.08.2020 (once every 4.7 days) using first sample dating (Binomial test, p = 0.002), or 40% more often using ML dating (p = 0.048; S2 Table in S1 Text).

For a fraction of phylogenetically inferred cross-border transmissions, it was possible to obtain independent confirmation. These were OBTs to countries with low overall number of cases (allowing cross-referencing of phylogenetic and epidemiological data) that sequence intensively; for example, New Zealand sequences and shares openly ~48% of its covid cases [17]. We were able to cross-reference five OBT events in this way, detailed below.

Two transmissions from Russia to New Zealand, giving rise to a total of 12 sequences (Fig 5; S7 Fig), were associated with mariners who traveled by a charter flight from Russia and Ukraine in the middle of October [18, 19]. A total of thirty-three cases in New Zealand were linked to this group, including two staff members of the isolation facility [20]. We show here that these mariners have brought with them at least two distinct variants of SARS-CoV-2. One of these variants, B.1.1.238, has been also independently introduced from Russia to Western Europe, and has been sampled in Denmark, England, Germany, Ireland, Netherlands, and Norway (Fig 5). This lineage carried the S:P681H mutation that was also present in Alpha, Kappa and Delta variants of concern and was shown to reduce recognition by some antibodies [21].

Additionally, a late June sequence from South Korea is nested within a Russian clade together with a number of UK sequences (S8A Fig), signifying an OBT. Unfortunately, no detailed information on the sources of the British sequences is available, but the sequence from South Korea is marked as imported in the GISAID metadata. We were not able to specify this further, although according to media reports, multiple ships with Russian mariners who tested positive for SARS-CoV-2 were located at the time in the port of Busan [22].

Finally, there are two more clusters of sequences from South Korea that group with Russian sequences (S8B and S8C Fig). The first includes one sequence dated October 14, and the second, two sequences dated October 30. All these sequences are marked as imported from Russia in GISAID metadata.

## Discussion

Availability of genetic sequences from different countries allows to infer international transmission of COVID-19 lineages. Here, we roughly doubled the amount of whole-genome

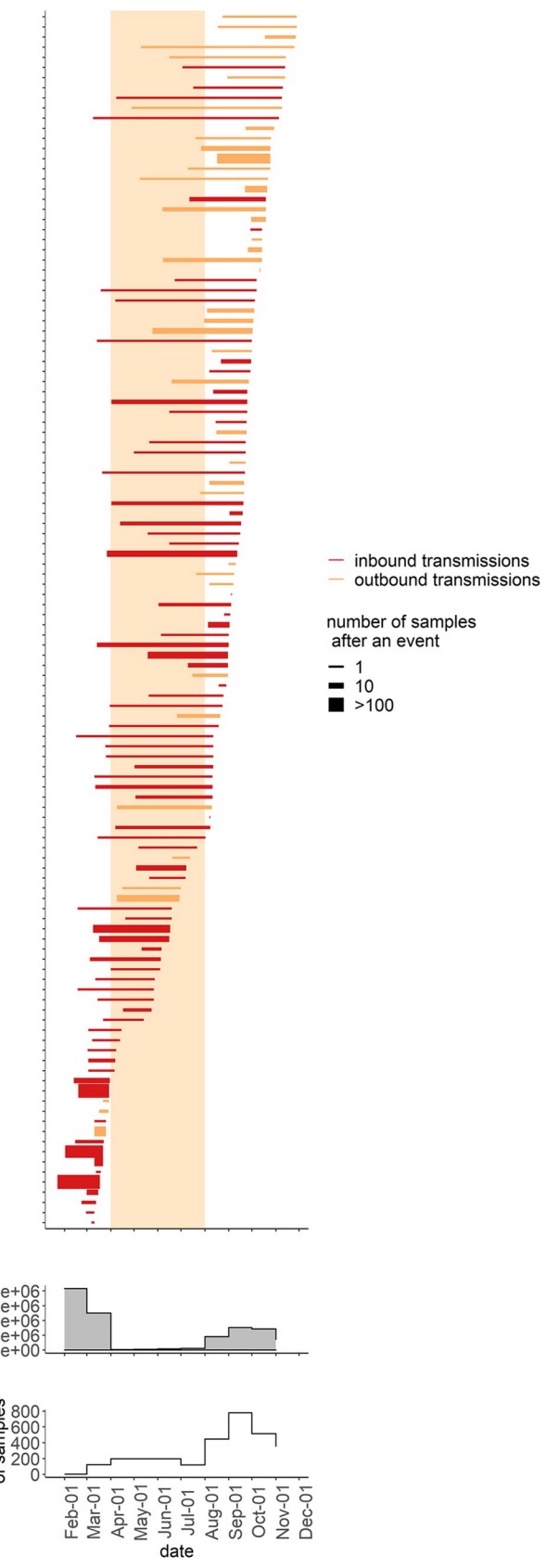

**Fig 4. Timeline for inbound and outbound transmissions of SARS-CoV-2 variants.** Bars correspond to individual IBTs (red) and OBTs (orange), with the left end of the bar corresponding to the date estimated for the corresponding node by the "node.dating" procedure [13] from the "ape" package of R language [14], and the right end of the bar corresponding to the date of the earliest Russian (for IBTs) or non-Russian (for OBTs) sample among the node's descendants. Line width identifies the number of sequences descendant to the IBT or OBT. Only cases with available sample dates are shown. Yellow background indicates the period of the most stringent closure of Russia's borders. Also shown is the number of international travellers going through Russian airports and the total number of Russian samples obtained in the corresponding month.

sequencing data for SARS-CoV-2 from Russia for 2020, giving us a high-resolution view of the onset and development of Russia's epidemic throughout its first year prior to the onset of the variants of concern.

Using genomic epidemiology, we show that genetically distinct SARS-CoV-2 variants have been imported into Russia at least 82 times, resulting in several hundreds of Russian transmission lineages. Similar to many other countries, we also detect outbound transmission of SARS-CoV-2 variants, and describe 43 putative outbound transmission events.

Our analysis has limitations. Similar to any phylogenetic analysis of viral transmission, it is limited by incompleteness and non-uniformity of sampling. The proportion of sequenced samples varies between countries, across locations within a country, across sampling methodologies (e.g., community sampling vs. individuals with travel history) and in time, which may bias the estimated number and direction of cross-border transmissions. Dating past events in the history of a lineage is also notoriously difficult, and prone with errors. Nonetheless, the strong phylogenetic clustering of sequences within a country (e.g. Fig 5) indicates that most inferred transmission lineages robustly reveal domestic transmission; and one of the two dating methods (based on the first sample) is independent of phylogenetic reconstruction.

Given these limitations, we observe intensive transmission of lineages both into Russia (IBTs) and outside it (OBTs). For a small fraction of cases, the direction of OBTs could be identified using public data. While these were the cases when the OBTs were stopped at the border, some of them resulted in onward transmission, highlighting the public health relevance of these events.

We find that border closure has led to a decrease in the number of cross-border transmissions. This decrease is unlikely to be artefactual. Indeed, during this period, increased global case counts should have increased the intensity of imports had the international passenger traffic remained the same. Moreover, at least two additional factors should have improved recognition of import events: increased sequencing, both within (Fig 4) and outside Russia; and increased viral genetic diversity, facilitating distinguishing between individual imports. For all these reasons, we expect an increase, rather than a decrease, in the number of transmissions in April-August 2020, compared to March 2020. Thus, the observed decrease in their number is likely a genuine effect of border closure.

Still, our analysis suggests that IBTs and OBTs of lineages were not fully stopped by a radical reduction of the air passenger flow (see S1 Table in S1 Text). While estimating the effect of the reduction in the numbers of IBTs and OBTs is beyond the scope of this study, modeling suggests that anything other than a near-complete border closure may have limited effect. Stochastic, spatially structured individual-based epidemiological simulations that examined the impact of interventions on clinical attack rates show that limiting the number of flights and/or canceling direct flights between two countries may not be sufficient to prevent transmission of variants between them [23]. Even weak passenger flow from a country with high prevalence of COVID-19 and high frequency of a variant may be sufficient for its import [16], and relatively few imports may suffice to cause sustainable domestic transmission, especially if these variants

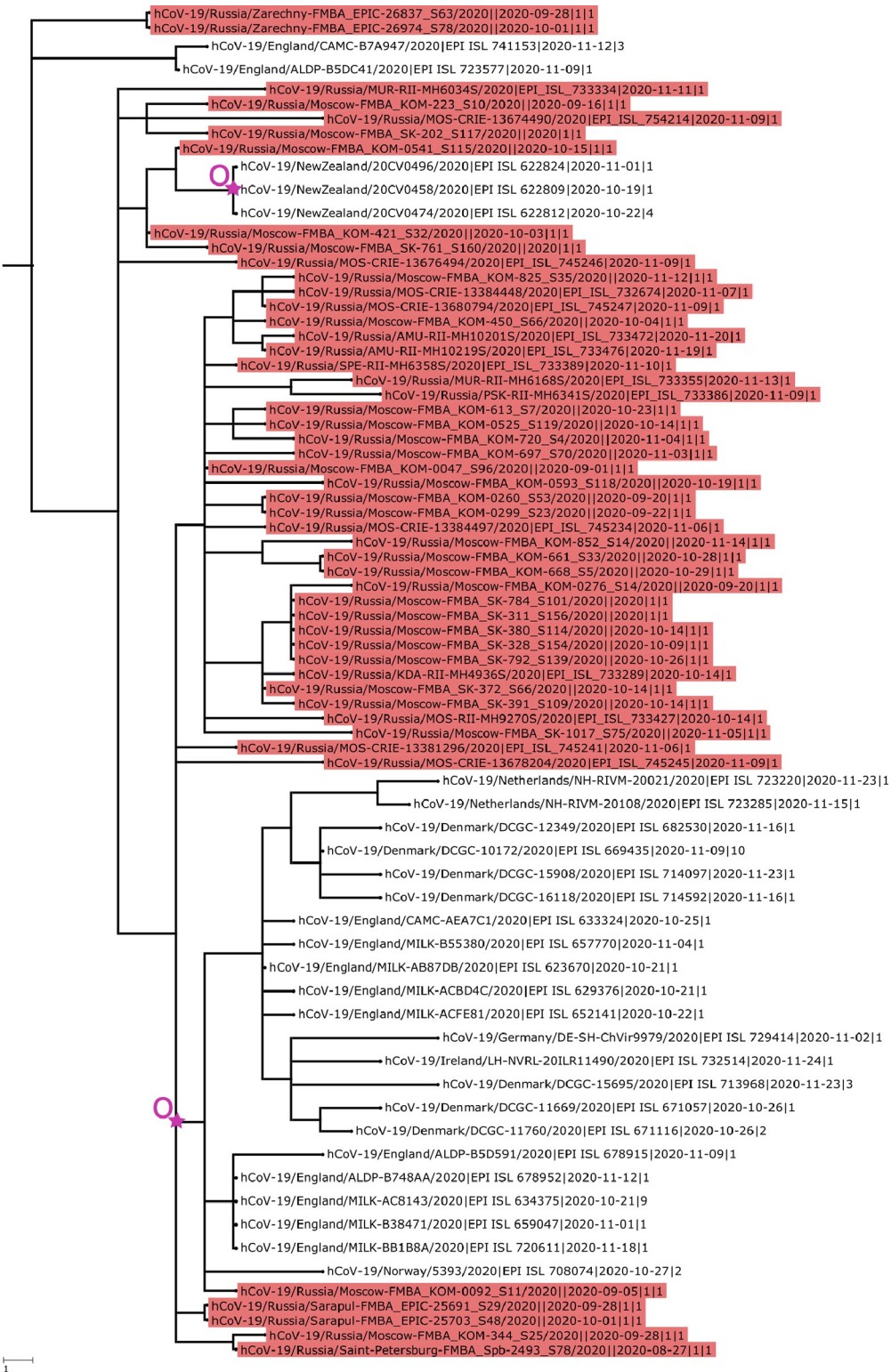

**Fig 5. IBTs and OBTs in the history of the B.1.1.238 lineage.** Branch lengths are measured in the number of changes. The number at the end of the sequence id represents the number of identical sequences (including the one shown) identified in the region on this date. Samples from Russia are identified with red labels. OBTs are indicated by a purple star and "O" letter.

confer selective advantage for the virus, e.g. due to increased transmissivity [23]. Subsequent emergence of multiple transmissivity-increasing variants calls for a systematic and effective approach to curbing their global spread.

## Materials and methods

### Sample collection

Nasopharyngeal swabs from patients with confirmed COVID-19 infection were collected between March 02 and November 25, 2020 by clinical hospitals subordinated to the Federal Medical-Biological Agency (FMBA) of Russia. All samples were collected to the transport media (Single-Use Virus Collection Tube, CDVCT-1, CDRICH) suitable for SARS-Cov-2 virus storage, transferred to the laboratory within 48 hours after collection and stored at -80˚C prior to RNA extraction and sequencing. Transportation and storage were compliant with the local normative documents for handling of biologically hazardous samples.

For each sample, the following data were stored: unique identifier, date of sample collection, and geographic region of origin. The obtained samples covered the following regions of Russia: Altai Region, Amur Region, Arkhangelsk Region, Astrakhan Region, Chelyabinsk Region, Chukotka Region, Chuvash Region, Irkutsk Region, Ivanovo Region, Kaluga Region, Kamchatka Region, Khabarovsk region, Khanty-Mansi Region, Kirov Region, Krasnodar Region, Krasnoyarsk Region, Kursk Region, Leningrad Region, Moscow, Moscow Region, Murmansk Region, Nizhny Novgorod Region, Novosibirsk Region, Penza Region, Perm Region, Primorsky Region, Rostov Region, Ryazan Region, Saint Petersburg, Sakha Region, Samara Region, Saratov Region, Smolensk Region, Stavropol Region, Sverdlovsk Region, Tomsk Region, Tver Region, Tyumen Region, Udmurtia Region, Ulyanovsk Region, Volgograd Region, Voronezh Region, and Zabaykalsky Region.

### Whole-genome sequencing

The presence of SARS-CoV-2 RNA was confirmed using the *AmpliTest* SARS-CoV-2 test kit (AmpliTest, Moscow, Russia), registration certificate № R3H 2020–9765 dated May 27, 2021, issued by the Federal Service for Surveillance in Healthcare. The sample preparation procedure was carried out in accordance with the manufacturer's instructions and included the isolation of RNA from biological material and RT-PCR with fluorescently labeled specific probes with real-time detection (40 amplification cycles in total). Samples showing a Ct greater than 25 were excluded from high-throughput sequencing analysis due to a library preparation success rate of less than 50%, and samples with Ct (Hex) below 25 were considered positive. RNA from positive samples was isolated using the Ribo-prep purification kit and reverse-transcribed using the Ampliseq cDNA Synthesis for Illumina kit (Illumina; San Diego, CA, USA)). The resulting cDNA was amplified using the AmpliSeq for Illumina SARS-CoV-2 Research Panel (Illumina) which contains 247 amplicons in 2 pools targeting the whole SARS-CoV-2 genome. Library preparation was performed using the AmpliSeq Library PLUS kit (Illumina). Library quality was assessed by capillary electrophoresis using the Agilent 2100 Bioanalyzer system (Agilent; Santa Clara, CA, USA). Library concentration was measured with the Qubit 4 Fluorometer (Thermo Fisher Scientific; Waltham, MA, USA) using the Qubit dsDNA HS Assay Kit (Thermo Fisher Scientific). Libraries with measured concentrations below 0.5 ng/μl and those that did not show the expected fragment length distribution were excluded from the analysis. Sequencing was carried out on the Illumina NextSeq 550 System with the NextSeq 500/550 Mid Output Kit v2.5 (300 Cycles) (Illumina). The manufacturers' recommendations were followed in all cases.

## Consensus calling

Paired-end sequencing data were generated for a total of 2,046 samples. After combining reads from the same samples, 1,903 unique sequenced samples remained. These sequences were then subjected to the following combination of read trimming and filtration: raw reads were trimmed with Trimmomatic-0.39 [24] to remove adapter sequences and low-quality ends. Trimmed reads were mapped onto the Wuhan-Hu-1 (MN908947.3) reference genome with bwa mem [25]. The following reads were then removed from the mapping: reads with abnormal insert length-to-read ratio (greater than 10 or less than 0.8); reads with insert length greater than 600; reads with more than 50% soft-clipped bases. 25 nucleotides were cropped from read ends using custom scripts to get rid of potential primer sequences. SNV and short indel calling was done with lofreq [26], with SNVs considered consensus if they were covered by at least 4 reads and supported by more than 50% of those reads; indels were considered consensus if they were covered by at least 20 reads with at least 50% of those supporting the variant; custom lofreq parameter settings were used to implement these thresholds. Regions that were covered by fewer than 2 reads or that were covered by 2 or 3 reads and not called as 100% reference were masked as N. If a locus was not in a masked region, but consensus calling thresholds were not achieved, the locus was called as reference. Sequences with more than 3,000 nucleotides marked as N were removed from further analysis, resulting in 1,636 sequences. Consensus sequences were generated by bcftools consensus [27].

## Data preparation and filtering

321,100 genomes of SARS-CoV-2 were downloaded from GISAID on January 6, 2021, (S2 Text) and aligned with MAFFT v7.453 [28] against the reference genome Wuhan-Hu-1/2019 (NCBI ID: MN908947.3) with—addfragments—keeplength options. 100 nucleotides from the beginning and from the end of the alignment were trimmed. After that, we excluded sequences (1) shorter than 29,000 bp, (2) with more than 3,000 (for Russian sequences) or 300 (for all other countries) positions of missing data (Ns), (3) excluded by Nextstrain [29], (4) from animals other than minks, (5) with a genetic distance to the reference genome more than four standard deviations from the epi-week mean genetic distance to the reference, or (6) obtained later than the most recent Russian sample (2020-11-28), leaving us with 218,889 sequences. The less stringent threshold in item (2) above used for the Russian sequences helped keep more data on Russian SARS-CoV-2 in the dataset. To this dataset, we added the 1,636 sequences produced in this study. For all countries but Russia, closely-related sequences were then collapsed within the country using cd-hit [30] with -c 0.99996 option. The final dataset comprised of 123,432 sequences was then annotated by the PANGOLIN package (v2.1.6, released on 2020-01-02). Following [16], we split the full, intractable dataset into five smaller subsets corresponding to the major distinct SARS-CoV-2 clades: A, B.x, B.1.x, B.1.GH and B.1.1 (see S9 Fig). We additionally masked a highly homoplasic site 11,083 prior to reconstructing phylogenies. GISAID names of sequenced samples are provided in S2 Text.

Number of daily new cases according to Johns Hopkins University (JHU) Center for Systems Science and Engineering (CSSE) for S1 Fig was taken from https://github.com/CSSEGISandData/COVID-19_Unified-Dataset [31].

## Phylogenetic analysis

For each of the five subsets, we constructed a phylogenetic tree using IQ-Tree v2.1.1 [32] under the GTR substitution model and '-fast -altr 1000' as options and reconstructed ancestral sequences at the internal tree nodes with TreeTime v0.8.0 [12]; this allowed us to convert branch lengths into the number of nucleotide changes. All subsequent analyses were

performed on each dataset independently. Sequences were grouped into Russian transmission lineages as described in [6], with the difference that we no longer distinguished between Russian transmission lineages and Russian stem-derived transmission lineages. Russian stem clusters, Russian singletons and Russian stem-derived singletons were inferred as described in [6].

Inbound transmissions (IBTs) and outbound transmissions (OBTs) were defined as follows. A node I is assumed to be ancestral to an IBT event if (i) all descendants of the node ancestral to I (other than I) are non-Russian; (ii) one or more of immediate descendants of I are clades consisting of only non-Russian sequences, and (iii) one or more of immediate descendants of I are Russian singletons or Russian transmission lineages (Fig 2C). Analogously, a node O is assumed to be ancestral to an OBT event if (i) all descendants of the node ancestral to O (other than O) are Russian; (ii) one or more of immediate descendants of O are Russian singletons or Russian transmission lineages; and (iii) one or more of immediate descendants of O are clades consisting of only non-Russian sequences (Fig 2D).

The above definitions are rather conservative and underestimate the true number of IBTs and OBTs. We additionally inferred migration events using Treetime v0.8.0 [12] (S6 Fig). We labeled every sequence on the tree as either Russian or non-Russian, and reconstructed the states of internal nodes for this trait. IBTs were then defined as edges whose parent and child nodes were inferred as non-Russian and Russian, respectively, with probabilities of at least 0.8. Groups of edges descendant to the same parent node were considered to constitute one IBT event. OBTs were defined analogously.

Data on monthly passenger traffic were obtained from the Federal Air Transport Agency webpage [9] and are provided in S1 Text.

Binomial test for uniformity of transmission events was performed with the "binom.test" function of R package "stats" [33]. Visualization was done in R with "Hmisc","tidyverse" and "ggpubr" packages [34–36].

## Validation of OBT events

Transmissions to New Zealand were cross-referenced with the local Christchurch cluster by sequence metadata (date and town of sample collection). No other cases were identified in Christchurch at that time. Information about transmissions to South Korea was extracted from GISAID metadata.

## Ethics statement

The study was presented to the Local Ethics Committee at the Centre of Hygiene and Epidemiology. The Committee concluded (protocol #03-04/20 of March 2, 2020) that the study does not make use of identifiable biological samples and does not bring forward any new sensitive data, since all samples were deidentified prior to receipt by the study team. Therefore, according to the rules of the Committee and national regulations, this project does not require ethical approval. Written informed consent was obtained from all study participants.

## Supporting information

**S1 Fig. The daily dynamic of new cases and sequenced cases in Russia throughout 2020.**
Yellow background indicates the period of the most stringent closure of Russia's borders.
(PNG)

**S2 Fig. Prevalence of major Pango lineages [10] in Russia (including the Republic of Crimea).** The data presented is for all sequences collected between March-November, 2020; see

S1 Fig for a breakdown by period. The circle size is proportional to the number of samples in corresponding regions, categorized by Pango lineages. Moscow is pooled with the surrounding Moscow Region, and Saint Petersburg is pooled with the surrounding Leningrad Region. Asterisks in Pango lineage designations correspond to pooled sets of lineages of that hierarchy level, except those listed in other categories; e.g., B.1.1.* includes B.1.1.7 but not B.1.1 or B.1.1.31.
(PNG)

**S3 Fig.** Prevalence of major PANGOLIN lineages [10] in Russia by period: A, March-July; B, August-November.
(PNG)

**S4 Fig. Rarefaction curves for numbers of inferred Russian transmission lineages.** The sequences from Moscow (A), Saint Petersburg (B) and all of Russia (C) were subsampled 10,000 times to the number shown on the horizontal axis, and the number of Russian transmission lineages was inferred. The shaded area shows the range of observed values.
(PNG)

**S5 Fig. Fractions of Russian transmission lineages in regions of Russia (including the Republic of Crimea) unique to one region or shared with other regions.** Circle size is proportional to the number of Russian transmission lineages in the corresponding region.
(PNG)

**S6 Fig. Timeline for inbound and outbound transmissions of SARS-CoV-2 variants, estimated by Treetime.** Notations are as in Fig 5.
(PNG)

**S7 Fig. OBTs to New Zealand.** Each tree represents an independent introduction event that occurred in the middle of October, each resulting in 6 sampled sequences. Branch lengths are measured in the number of changes. Samples from Russia are identified with red labels. OBTs are indicated by purple star and "O" letter. The number at the end of the sequence id represents the number of identical sequences (including the one shown) identified in the region on this date.
(PNG)

**S8 Fig. OBTs to South Korea.** Notation as in S5 Fig.
(PNG)

**S9 Fig. Schematic representation of five distinct SARS-CoV-2 clades analysed independently in the work.** B.x consists of the B lineage and all its descendants except B.1 and lineages descendant from it; similarly, B.1.x includes B.1 and all its descendants but B.1.1 lineage and GH clade (GISAID nomenclature; denoted here as B.1.GH); B.1.1 and B.1.GH clades are analyzed separately. Sites carrying the key mutations defining the specified clades are indicated.
(PNG)

**S1 Text.**
(DOCX)

**S2 Text.**
(ZIP)

**S3 Text.**
(TXT)

## Acknowledgments

We thank Sergei L Kosakovsky Pond for help with HyPhy analyses, and Evgeniya Alekseeva and members of the Bazykin lab for fruitful discussions. We thank all of the authors who have contributed genome data on GISAID (see S2 Text for the list).

## Author Contributions

**Conceptualization:** Galya V. Klink, Ksenia R. Safina, Sofya K. Garushyants, Georgii A. Bazykin.

**Formal analysis:** Galya V. Klink, Ksenia R. Safina, Georgii A. Bazykin.

**Funding acquisition:** Georgii A. Bazykin, German A. Shipulin, Sergey M. Yudin, Veronika I. Skvortsova.

**Investigation:** Alina Matsvay, Galya V. Klink, Ksenia R. Safina, Elena Nabieva, Dmitry Biba, Georgii A. Bazykin, Ivan M. Mikhaylov, Anna V. Say, Anastasiya I. Zakamornaya, Anastasiya O. Khakhina, Tatiana S. Lisitsa, Andrey A. Ayginin, Ivan S. Abramov, Sergey A. Bogdan, Kseniya B. Kolbutova, Daria U. Oleynikova, Tatiana F. Avdeenko.

**Resources:** German A. Shipulin, Sergey M. Yudin, Veronika I. Skvortsova.

**Supervision:** Georgii A. Bazykin, German A. Shipulin, Sergey M. Yudin, Veronika I. Skvortsova.

**Visualization:** Galya V. Klink, Ksenia R. Safina, Sofya K. Garushyants.

**Writing – original draft:** Galya V. Klink, Georgii A. Bazykin.

**Writing – review & editing:** Galya V. Klink, Ksenia R. Safina, Elena Nabieva, Georgii A. Bazykin.

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
