## [Decision Letter · Decision Letter 0]

16 Aug 2022

PONE-D-22-17948Genomic epidemiology of SARS-CoV-2 in Russia reveals recurring cross-border transmission throughout 2020PLOS ONE

Dear Dr. Klink,

Thank you for submitting your manuscript to PLOS ONE. After careful consideration, we feel that it has merit but does not fully meet PLOS ONE’s publication criteria as it currently stands. Therefore, we invite you to submit a revised version of the manuscript that addresses the points raised during the review process.

We look forward to receiving your revised manuscript.

Kind regards,

Mohd Adnan, PhD

Academic Editor

PLOS ONE

Journal Requirements:

"This study was partially supported by RFBR project 20-04-60556. The funders had no role in study design, data collection and analysis, decision to publish, or preparation of the manuscript".

5. We note that Figures 1, 4 and S1 in your submission contain map/satellite images which may be copyrighted. All PLOS content is published under the Creative Commons Attribution License (CC BY 4.0), which means that the manuscript, images, and Supporting Information files will be freely available online, and any third party is permitted to access, download, copy, distribute, and use these materials in any way, even commercially, with proper attribution. For these reasons, we cannot publish previously copyrighted maps or satellite images created using proprietary data, such as Google software (Google Maps, Street View, and Earth). For more information, see our copyright guidelines: http://journals.plos.org/plosone/s/licenses-and-copyright.

a) You may seek permission from the original copyright holder of Figures 1, 4 and S1 to publish the content specifically under the CC BY 4.0 license.  

Additional Editor Comments:

Manuscript does not fulfill the standards established for the journal to be considered for publication in its current form. I agree with the reviewers that manuscript requires substantial revision and additional work to support the conclusion and improve the quality of the publication. Please see the detailed comments made by the reviewers.

Reviewers' comments:

Reviewer's Responses to Questions

**Comments to the Author**

1. Is the manuscript technically sound, and do the data support the conclusions?

Reviewer #1: Partly

Reviewer #2: Yes

Reviewer #3: Partly

Reviewer #4: Yes

Reviewer #5: Yes

2. Has the statistical analysis been performed appropriately and rigorously? 

Reviewer #1: N/A

Reviewer #2: Yes

Reviewer #3: N/A

Reviewer #4: Yes

Reviewer #5: Yes

3. Have the authors made all data underlying the findings in their manuscript fully available?

Reviewer #1: Yes

Reviewer #2: Yes

Reviewer #3: Yes

Reviewer #4: Yes

Reviewer #5: Yes

4. Is the manuscript presented in an intelligible fashion and written in standard English?

Reviewer #1: Yes

Reviewer #2: Yes

Reviewer #3: Yes

Reviewer #4: No

Reviewer #5: Yes

5. Review Comments to the Author

Reviewer #1: I would recommend addressing the following issues before it can be considered for publication on this journal.

Major revisions:

General comments: I’m concerned about the heterologous stringency for sequence selection, since up to 10% Ns were allowed in Russian sequences whilst non-Russian sequences were required to have far fewer (1%). The former are actually quite low, so I fear this may impact phylogenetic reconstruction.

Also, only by accessing figure S2 and the methods did the IBT and OBT become clear. As these were central to the study, they should be clearly defined earlier in the results. The reader should be provided a clear explanation on why the authors assume inbound or outbound transmission and why is it relevant.

Also, figures are almost peripheric to the main manuscript, they are not really used for support. They should be linked to the main study and discussed accordingly.

Ultimately, the manuscript could use a better explanation on the importance of this analysis and why was it limited to the first year? If the answer is “demonstrating that transfrontier transmission was ongoing throughout 2020”, then why stop there and how can the authors be sure of the actual direction since sequenced genomes represent only a very small percentage of total confirmed cases?

Finally, no statistical comparisons were shown at all, which could prove useful to determine differences within regions or countries of origin. Why was nothing compared, was there not enough data?

Introduction section

L47: The authors should provide a figure depicting total daily cases in Russia with a sliding average window (7 or 14-day avg) including 2020 (perhaps as a supplementary figure) for anyone who is not familiar with Russia’s situation during the pandemic. The figure should also include the total sequenced genomes (different y axis, perhaps) from the same time period as the first part tend to have very few samples. Finally, it would be rather useful if the authors could mark up some of the main events that are relevant for the current study (first documented case, border closure and opening).

L69: Please briefly specify how were non-Russian sequences selected (randomized, preselected, etc.) and perhaps reference the methods section here as well.

Results section:

Figure 1: The authors should include international borders in the map as land frontiers are important determinants of the actual dynamics.

Figure 2: Nothing is said in the main text about this figure, even though it is quite interesting. Also, perhaps the authors should include different y axes as to clearly define in which periods were these most prevalent. Also, area plots tend to be misleading when only few samples are available, as may be the case with the first introductions. It may be a good idea to include stacked area plots (not as frequencies but as raw totals, maybe in a different supplementary figure).

Also, consider moving this to supplementary and bringing up S2 to the main figures as it makes the next part easier to understand.

L70-L77: I found this paragraph to be somewhat confusing, perhaps it could be improved by using some more precise terms:

-L70: First, please refer to sequences, rather than samples, to avoid confusion, or make sure the reader understands they are used interchangeably here. Also, please be clear as to what the authors mean by “Russian transmission lineage” (were all of them monophyletic clades containing only sequences from Russian samples? Were there any non-Russian sequences in Russian transmission lineages? To what extent?). Also, other clade limits are not entirely clear in the text. (From Fig S2 it is clear that, apart from transmission lineages, others were actually paraphyletic) but, again, was there a minimum quota in the prevalence of Russian sequences within?

-L71: The term “Phylogenetic positions” on its own may be mistaken with actual mutations per se. Maybe use “On the basis of their position in the phylogeny (ML tree)”?

-L72: How were “characteristic changes” defined and according to what? “At its base” refers to the actual last common ancestor (LCA) of a monophyletic clade. Please use standard phylogenetic terms to avoid confusion (e.g. monophyletic clades having x mutations in common).

-L73: Variants, rather than strains (as of 2022, no serotypes nor strains have been defined officially).

-L74: Rather, these were phylogenetically closer to non-Russian clades than to other Russian sequences. It would be better if the authors used standard evolutionary terms instead.

L81: How can the authors be sure that these were actually imported and not the other way around? Can they confirm community transmission from these data? What if some of these remained undetected even though they were in Russia first?

Figure 3: “gray circles” also refer to lineages A* and B* according to the legend, so please be careful in the description. Also do the gray circles with dashed lines refer to actual sequences within the same clades that were also found in Russia earlier as basal/stem nodes in the tree (this is what I get from the figure description) or could these also be from non-Russian basal sequences as I think they do? This wasn’t clear to me but would explain them closely resembling other international items. Also, what are those four gray lines with no color terminal nodes on top of the figure (first 4 lines)?

L84: Rarefactions with no plateau do not necessarily mean that they were constantly imported (even though that might be a potential explanation) but only that some were missing due to sampling, so be careful when addressing this in the manuscript.

L89: Were bordering regions more prone to exchange variants? Also, were there variants exclusively seen in Russia or heavily present compared to the world during this period? It would be useful if the authors could provide some table containing those in a much higher prevalence in Russia than in other parts of the world.

L95: What about clades having sequences from both Russian and non-Russian origin in outgroups, were these discarded for this estimation?

L100: Could the authors very briefly describe how Treetime (also ML) differs from their method from a technical perspective?

L107: Were there any differences during border closure?

L110: Each of these particular cases should include some information on the actual variants they belonged to and why they are actually relevant for the study.

L121: Adding upon the last comment, which other variants are exemplified here and why was this particular mutation (S:P681H) relevant?

L124: What were the variants in this clade? Why was their relevance?

L131: What about these two clusters, why are they mentioned?

Discussion

L140: This substantial increase is with respect to existing sequences (how many were there?)? I’m assuming these were the ~1600 sequences mentioned above? The authors could mention them explicitly here as it is an important result.

L142: The authors should further discuss the rationale behind inbound and outbound transmissions in this section (how should outgroups be defined in order to determine a clade may be an IBT/OBT). Also, IBTs and OBT topologies could arise from undersampled regions and other contextual artifacts and this should be included here. How can the authors be sure of the actual direction of the transmission (or else mention these as limitations)?

L149: What are some examples of other countries? Since the percentage of crossing passengers that are sampled is always low, probability of picking out the exact event is almost nonexistent. Instead, this might get sequenced by chance but only after they are already rather common. Ultimately, I don’t think these events in particular may be important, but their dynamics are.

L153: Please elaborate on which epidemiological simulations the authors refer to as plenty have been shown before (just what kind of simultation and what variables are considered).

L161: Will the authors continue this study or expand it beyond 2020? What are the current perspectives?

References:

Ref 5 was not working. Please check all others. When were they last accessed?

L63: When the authors mention the Diamond Princess patient, they should include a reference for context.

Materials and Methods:

L204: As a recommendation for further studies, cutadapt may be used to trim the adapters more finely so there is no need to trim 25 nt from each end, which may improve mapping.

L214: Just another recommendation: Consensus by bcftools may be upgraded with ivar,to determine which variants are most common. This can be used to reduce Ns where polymorphic positions are found but not prevalent.

L220-221: These heterologous filters (~10% Ns for Russian items and 1% for non-Russian ones) may lead to issues in the downstream analyses and artifacts in phylogenetic trees due to some positions being missing from the alignment. I understand if this was selected as to avoid losing data but the consensus may be improved to prevent this by a major vote rule over SNP positions instead (ivar can be used to work this out).

L222: Why were minks left in the study?

Minor revisions

L47: Why January 19, 2021? It would be better if the authors reported a more recent figure, even if the study is limited to 2020.

L49: “from four-five million” Are these averages? Could the authors provide a reference for this?

L60: March 2.

L76: Unique mutations instead of characteristic changes or else describe what are those.

L80: All were sequenced from samples, maybe specify them as coming from the original Russian set.

L104: Please specify that inferred IBTs and OBTs refer to the the former 82/43.

L148: case*.

Reviewer #2: This is a well-written manuscript about cross-border transmissions of SARS-CoV2 variants. The authors attempted to establish a link between travel restrictions and the transmission of variants in Russia. Although, it is important to note that travel restrictions were mainly imposed broadly to contain the spread of the SARS-CoV-2 virus rather than variants. It is challenging for our increasingly interdependent world to impose complete border closure. For example, the presence of the SARS-CoV-2 virus in water bodies shared by countries and the movement of sub-clinically infected animals (White-tailed deer) can also contribute to the transmission of new variants. That said, the authors have provided enough evidence to confirm their findings. Please address these minor comments,

1. Please include the limitations to this study.

2. Line 149: “ ..corresponding countries,” What are these countries?

3. Line 153: “ radical reduction of the air passenger flow.” please refer to results.

4. I believe it may be Federal Air transport Agency webpage is not accessible outside Russia, at least, it did not work for me. Therefore, please provide the data used for analysis as supplementary information

5. Line 183: “ The presence of SARS-CoV-2 RNA…” please provide manufacturer information and test protocol/ reaction conditions for RT-qPCR. Why were Ct below 25 considered positive, explain? How many cycles were the samples run?

6. What was the quality and quantity of RNA used for WGS? Please include the information in the manuscript.

7. There were several references to websites and news outlets, some of which are local websites. It is important to include the date and time they were accessed.

Reviewer #3: The manuscript appears to have been sent for publication very late. Actually, there are much information about the epidemiology of SARS-CoV2 . Thus the results of this work -in my opinion -are obsolete and lack of novelty.

Reviewer #4: Some of the grammar can be improved. An email with comments is to be sent.

The paper is well written and understandable and the methodology has given all tools used so should be replicated if one desires to.

The paper is a bit technical in analysis but that is expected because of the analysis tools available.

A revision of the grammar should allow the paper to be published

In the discussion, the author should compare IBTs and OBTs introduced during the travel ban versus when the ban was lifted to be able to clearly conclude that there was no major impact.

.

Reviewer #5: The article: "Genomic epidemiology of SARS-CoV-2 in Russia reveals

recurring cross-border transmission throughout 2020." explains in an organized way

the effect of border closures on the transmission of SARS-COV2 infection during the first year of the pandemic.

This information is relevant because it can serve as a guiding element in decision-making by the authorities.

in health in the face of the eminent threat of a rapidly transmitted respiratory disease, as it turned out to be human coronaviruses.

6. PLOS authors have the option to publish the peer review history of their article (what does this mean?). If published, this will include your full peer review and any attached files.

Reviewer #1: No

Reviewer #2: No

Reviewer #3: No

Reviewer #4: **Yes: **Rachel Achilla

Reviewer #5: No

---

## [Author Response · Author response to Decision Letter 0]

7 Feb 2023

Reviewer #1: I would recommend addressing the following issues before it can be considered for publication on this journal.

Major revisions:

General comments: I’m concerned about the heterologous stringency for sequence selection, since up to 10% Ns were allowed in Russian sequences whilst non-Russian sequences were required to have far fewer (1%). The former are actually quite low, so I fear this may impact phylogenetic reconstruction.

We understand your concerns. However, the viral genome is long enough, and even with 10% missing, poorly covered sequences still have ~27k bp for phylogenetic inference. The logic for the different thresholds was that as we were focused on the situation in Russia, we were interested in keeping as many Russian samples as possible, while non-Russian samples were abundant (to the extent that they had to be subsampled) and therefore could be filtered more stringently. 

Also, only by accessing figure S2 and the methods did the IBT and OBT become clear. As these were central to the study, they should be clearly defined earlier in the results. The reader should be provided a clear explanation on why the authors assume inbound or outbound transmission and why is it relevant.

We now expand the motivation for the study and explain the logic behind the definitions of IBTs and OBTs in more detail in the main text. We also moved Fig. S2 to the main text (it is Fig. 2 now). 

Also, figures are almost peripheric to the main manuscript, they are not really used for support. They should be linked to the main study and discussed accordingly.

We now expand the discussion of Fig. 2 (which is Fig. 1 now) and move Fig. 1 and Fig. 4 to Supplements (now Fig. S2 and Fig. S5, respectively).

Ultimately, the manuscript could use a better explanation on the importance of this analysis and why was it limited to the first year? If the answer is “demonstrating that transfrontier transmission was ongoing throughout 2020”, then why stop there and how can the authors be sure of the actual direction since sequenced genomes represent only a very small percentage of total confirmed cases?

We used all sequencing data available by the time of this study. The studied time span includes the period of the most stringent lockdown in Russia together with the preceding and subsequent periods.

The actual direction of cross-border transmissions was found with the help of a phylogenetic tree. To be confident in the direction of transmission, for imports into Russia we asked a purely Russian clade to be descended from and adjacent to purely non-Russian clades. Analogously, for exports from Russia we asked a purely non-Russian clade to be descended from and adjacent to purely Russian clades. 

Finally, no statistical comparisons were shown at all, which could prove useful to determine differences within regions or countries of origin. Why was nothing compared, was there not enough data?

There were not enough cross-border transmissions, and too many source countries, for a meaningful statistical analysis of whether some countries were more likely to transmit into Russia than others. We now add a formal statistical assessment of the effect of border closure (see below).

Introduction section

L47: The authors should provide a figure depicting total daily cases in Russia with a sliding average window (7 or 14-day avg) including 2020 (perhaps as a supplementary figure) for anyone who is not familiar with Russia’s situation during the pandemic. The figure should also include the total sequenced genomes (different y axis, perhaps) from the same time period as the first part tend to have very few samples. Finally, it would be rather useful if the authors could mark up some of the main events that are relevant for the current study (first documented case, border closure and opening).

While the number of samples is presented as the lowest panel in Fig. 5, we now also provide it in more detail (smaller bins) in Fig. S1, together with the case counts for the corresponding dates. A period of the most stringent travel ban in Russia is shown by the yellow square on this figure.

L69: Please briefly specify how were non-Russian sequences selected (randomized, preselected, etc.) and perhaps reference the methods section here as well.

We now reference the Methods section in the text. Details on GISAID filtering are perhaps too technical for this section, and their absence doesn’t affect understanding.

Results section:

Figure 1: The authors should include international borders in the map as land frontiers are important determinants of the actual dynamics.

We disagree: the land frontiers have had little contribution to lineage composition in this pandemic, at least in Russia. For example, Russia has a 4209-km land border with China, but we have previously shown (Komissarov et al. 2021) that although SARS-CoV-2 originated in China, just one of the 67 earliest introductions in February-March 2020 has been from China. While Russia has land borders with 14 countries, the earliest transmission lineages were apparently imported from the USA, Chile, England, France, and Denmark (Komissarov et al. 2021) - all countries with no terrestrial border with Russia, so the bulk of the virus has been probably imported by plane or sea. 

Figure 2: Nothing is said in the main text about this figure, even though it is quite interesting. Also, perhaps the authors should include different y axes as to clearly define in which periods were these most prevalent. Also, area plots tend to be misleading when only a few samples are available, as may be the case with the first introductions. It may be a good idea to include stacked area plots (not as frequencies but as raw totals, maybe in a different supplementary figure).

We now added a paragraph to the Results discussing this figure. The current version of the figure presents both the numbers (left) and fractions (right) of Pango lineages, thus showing the data requested by the Reviewer (i.e., when the lineages were the most prevalent and how robust the frequency estimates are). 

Also, consider moving this to supplementary and bringing up S2 to the main figures as it makes the next part easier to understand.

We now moved Fig. S2 to the main text.

L70-L77: I found this paragraph to be somewhat confusing, perhaps it could be improved by using some more precise terms:

We rephrased this paragraph now.

-L70: First, please refer to sequences, rather than samples, to avoid confusion, or make sure the reader understands they are used interchangeably here. 

Thank you, we have changed “samples” to “sequences” throughout the manuscript wherever appropriate. 

Also, please be clear as to what the authors mean by “Russian transmission lineage” (were all of them monophyletic clades containing only sequences from Russian samples? Were there any non-Russian sequences in Russian transmission lineages? To what extent?). Also, other clade limits are not entirely clear in the text. (From Fig S2 it is clear that, apart from transmission lineages, others were actually paraphyletic) but, again, was there a minimum quota in the prevalence of Russian sequences within?

Following previous work by ourselves and others, we defined Russian transmission lineages as monophyletic groups (clades) carrying more than one sequence all of which were Russian. We now give this definition early on, and move Fig. S2 illustrating it to the main text.

-L71: The term “Phylogenetic positions” on its own may be mistaken with actual mutations per se. Maybe use “On the basis of their position in the phylogeny (ML tree)”?

Changed as suggested.

-L72: How were “characteristic changes” defined and according to what? “At its base” refers to the actual last common ancestor (LCA) of a monophyletic clade. Please use standard phylogenetic terms to avoid confusion (e.g. monophyletic clades having x mutations in common).

We now clarified the definitions as suggested. The reviewer is right: “characteristic changes” were just common mutations, now explained.

-L73: Variants, rather than strains (as of 2022, no serotypes nor strains have been defined officially).

Changed as suggested.

-L74: Rather, these were phylogenetically closer to non-Russian clades than to other Russian sequences. It would be better if the authors used standard evolutionary terms instead.

We reformulated the definition of Russian transmission lineages now.

L81: How can the authors be sure that these were actually imported and not the other way around? Can they confirm community transmission from these data? What if some of these remained undetected even though they were in Russia first?

We agree that new Russian transmission lineages can appear due to imports as well as due to export events. We now changed the phrase “were brought into Russia repeatedly” to “crossed the Russian border repeatedly”. 

Figure 3: “gray circles” also refer to lineages A* and B* according to the legend, so please be careful in the description. Also do the gray circles with dashed lines refer to actual sequences within the same clades that were also found in Russia earlier as basal/stem nodes in the tree (this is what I get from the figure description) or could these also be from non-Russian basal sequences as I think they do? This wasn’t clear to me but would explain them closely resembling other international items. Also, what are those four gray lines with no color terminal nodes on top of the figure (first 4 lines)?

We now made the description more clear.

The gray circles with dashed lines represented the date of the earliest Russian sample on the stem of the transmission lineage. If there were no Russian sequences on the stem, gray circle and dashed line were not drawn. 

Gray lines without colored terminal nodes are transmission lineages without dated sequences. It is now mentioned in figure legend. 

L84: Rarefactions with no plateau do not necessarily mean that they were constantly imported (even though that might be a potential explanation) but only that some were missing due to sampling, so be careful when addressing this in the manuscript.

The current version of our phrase mentions both these possibilities: “...suggesting that some lineages remained undetected and/or that lineages continued to be imported”.

L89: Were bordering regions more prone to exchange variants? Also, were there variants exclusively seen in Russia or heavily present compared to the world during this period? It would be useful if the authors could provide some table containing those with a much higher prevalence in Russia than in other parts of the world.

As discussed above, most imports (and exports) were probably by air or sea rather than by land. We have described the relative prevalence of lineages within and outside Russia previously (Klink et al. 2021 https://journals.plos.org/plosone/article?id=10.1371/journal.pone.0270717); we now also added a paragraph in Results discussing this.

L95: What about clades having sequences from both Russian and non-Russian origin in outgroups, were these discarded for this estimation?

Such cases were not considered as imports or exports, allowing us to obtain a conservative estimation of the number of transmission events. 

L100: Could the authors very briefly describe how Treetime (also ML) differs from their method from a technical perspective?

Our way of defining OBTs and IBTs is based solely on our conservative definition of what we consider as cross-border transmissions along the phylogenetic tree. This is clarified now. As we discuss, our parsimony-based method is more conservative, focusing on the most reliable trans-border transmissions. 

L107: Were there any differences during border closure?

We now analyze these differences explicitly (see response to Reviewer 4).

L110: Each of these particular cases should include some information on the actual variants they belonged to and why they are actually relevant for the study.

The cases that we described in the manuscript represent cross-border transmission events that we detected using our phylogeny-based approach and that we were able to confirm from independent sources of information, such as media reports. These cases are relevant to the study because they confirm the relevance of using phylogenetic tree to find cross-border transmissions. 

L121: Adding upon the last comment, which other variants are exemplified here and why was this particular mutation (S:P681H) relevant?

See comment to L110. S:P681H is a famous spike mutation that confers resistance to antibodies and is involved in several SARS-CoV-2 variants of concern which evolved later; this is now mentioned.

L124: What were the variants in this clade? Why was their relevance?

See comment to L110

L131: What about these two clusters, why are they mentioned?

See comment to L110

Discussion

L140: This substantial increase is with respect to existing sequences (how many were there?)? I’m assuming these were the ~1600 sequences mentioned above? The authors could mention them explicitly here as it is an important result.

We rephrase this to read “roughly double”. The numbers implied are indeed the ones provided in the beginning of the Results section: 1636 sequences obtained in this study, vs. 1570 obtained in other work.

L142: The authors should further discuss the rationale behind inbound and outbound transmissions in this section (how should outgroups be defined in order to determine a clade may be an IBT/OBT). Also, IBTs and OBT topologies could arise from undersampled regions and other contextual artifacts and this should be included here. How can the authors be sure of the actual direction of the transmission (or else mention these as limitations)?

We now include a limitations section in Discussion where we mention the possible effect of non-uniform sampling.

L149: What are some examples of other countries? Since the percentage of crossing passengers that are sampled is always low, probability of picking out the exact event is almost nonexistent. Instead, this might get sequenced by chance but only after they are already rather common. Ultimately, I don’t think these events in particular may be important, but their dynamics are.

We now provide a list of these countries for lineage B.1.1.238 from Fig. 5 above in the Results. We agree that we are likely to miss many, and probably most, of the transmission events. 

L153: Please elaborate on which epidemiological simulations the authors refer to as plenty have been shown before (just what kind of simultation and what variables are considered).

We meant stochastic, spatially structured individual-based simulations used in “Strategies for mitigating an influenza pandemic” (Ferguson et al. 2006) paper. We now added their short description.

L161: Will the authors continue this study or expand it beyond 2020? What are the current perspectives?

The aim of this study was to follow the beginning of the epidemic of SARS-CoV2 in Russia and to trace the intensity of cross-border transmissions of the infection at that time. We focus on the period prior to the spread of variants of concern; over this period, lineage dynamics has been determined mostly by the epidemiology rather than differences in fitness between variants, facilitating analysis.

We already described the subsequent dynamics of the epidemic in Russia in other works (Klink et al. 2021a,b; omicron). We do plan to monitor lineage dynamics in Russia further.

References:

Ref 5 was not working. Please check all others. When were they last accessed?

We checked all links from the Reference list. Refs 18 and 19 were outdated, we now updated them. All other refs were actual, including Ref 5.

L63: When the authors mention the Diamond Princess patient, they should include a reference for context.

We now included the link on the paper about this case. 

Materials and Methods:

L204: As a recommendation for further studies, cutadapt may be used to trim the adapters more finely so there is no need to trim 25 nt from each end, which may improve mapping.

Thank you!

L214: Just another recommendation: Consensus by bcftools may be upgraded with ivar,to determine which variants are most common. This can be used to reduce Ns where polymorphic positions are found but not prevalent.

Thank you for this recommendation, as well! We decided to make our procedure as conservative as possible, but calling ambiguous nucleotides is certainly a good idea.

L220-221: These heterologous filters (~10% Ns for Russian items and 1% for non-Russian ones) may lead to issues in the downstream analyses and artifacts in phylogenetic trees due to some positions being missing from the alignment. I understand if this was selected as to avoid losing data but the consensus may be improved to prevent this by a major vote rule over SNP positions instead (ivar can be used to work this out).

Multiple N in Russian samples usually come from regions of extremely low or no coverage. Using ivar sample-wise wouldn’t solve the issue with multiple Ns. Substituting Ns with alignment consensus is not necessarily a good idea either as wrong inference would create a false substitution event, so we trusted IQTree to fairly treat missing data.

L222: Why were minks left in the study?

For historic reasons. There were no mink sequences from Russia, so this did not affect our results.

Minor revisions

L47: Why January 19, 2021? It would be better if the authors reported a more recent figure, even if the study is limited to 2020.

This is when this study has been performed. Ref 5 which this sentence refers to contains a link on a regularly updated server, therefore, anyone who is interested in the current figure might see it. 

L49: “from four-five million” Are these averages? Could the authors provide a reference for this?

These are approximate numbers, as the actual numbers are 4929170 for January and 4166359 for February. The reference for this data is provided in the end of the sentence (Ref.9), and now the data itself is also available in Supplementary File 3. 

L60: March 2.

Fixed.

L76: Unique mutations instead of characteristic changes or else describe what are those.

By “characteristic mutations” we meant mutations that occurred on the ancestral branch of the transmission lineage.We now changed “characteristic changes at its base” to “mutations leading to its LCA”.

L80: All were sequenced from samples, maybe specify them as coming from the original Russian set.

We reformulated this paragraph to make it more understandable now.

L104: Please specify that inferred IBTs and OBTs refer to the the former 82/43.

Transmission events that are found by Treetime with Maximum Likelihood approach included 94% of events that were found by our definition (82/43). This is mentioned below in the same paragraph.

L148: case*.

We now changed “samples” to “cases”, thank you.

Reviewer #2: This is a well-written manuscript about cross-border transmissions of SARS-CoV2 variants. The authors attempted to establish a link between travel restrictions and the transmission of variants in Russia. Although, it is important to note that travel restrictions were mainly imposed broadly to contain the spread of the SARS-CoV-2 virus rather than variants. It is challenging for our increasingly interdependent world to impose complete border closure. For example, the presence of the SARS-CoV-2 virus in water bodies shared by countries and the movement of sub-clinically infected animals (White-tailed deer) can also contribute to the transmission of new variants. That said, the authors have provided enough evidence to confirm their findings. Please address these minor comments,

We thank the Reviewer for understanding the importance of such a type of work and for appreciating our research.

1. Please include the limitations to this study.

We now include limitations in the Discussion section.

2. Line 149: “ ..corresponding countries,” What are these countries?

Here, we meant one of OBT events that were shown on Figure 6. For this case, the virus entered Denmark, Germany, Netherlands, Norway and the UK, according to the phylogenetic tree.

3. Line 153: “ radical reduction of the air passenger flow.” please refer to results.

We now refer to Supplementary File 3 that contains the information about international air passenger flow for Russia for 2020.

4. I believe it may be Federal Air transport Agency webpage is not accessible outside Russia, at least, it did not work for me. Therefore, please provide the data used for analysis as supplementary information

This data is now provided in Supplementary File 1.

5. Line 183: “ The presence of SARS-CoV-2 RNA…” please provide manufacturer information and test protocol/ reaction conditions for RT-qPCR. Why were Ct below 25 considered positive, explain? How many cycles were the samples run?

RT-qPCR was performed with AmpliTest SARS-CoV-2 test kit (AmpliTest, Moscow, Russia), registration certificate № РЗН 2020-9765 dated May 27, 2021, issued by the Federal Service for Surveillance in Healthcare. The sample preparation procedure was carried out in accordance with the manufacturer's instructions and included the isolation of RNA from biological material and RT-PCR with fluorescently labeled specific probes with real-time detection (40 amplification cycles in total). Samples showing a Ct greater than 25 were excluded from high-throughput sequencing analysis due to a library preparation success rate of less than 50%. This information is now included in the manuscript.

6. What was the quality and quantity of RNA used for WGS? Please include the information in the manuscript.

Direct assessment of the quality and quantity of RNA was not carried out. We performed sample preparation of samples for which the Ct(Hex) obtained by the AmpliTest SARS-CoV-2 test kit (AmpliTest, Moscow, Russia) was below 25. Prepared libraries with measured concentrations below 0.5 ng/µl and those that did not show the expected fragment length distribution were excluded from the analysis. We now included more details of the quality assessment in the manuscript.

7. There were several references to websites and news outlets, some of which are local websites. It is important to include the date and time they were accessed.

We added dates of the last accession now.

Reviewer #3: The manuscript appears to have been sent for publication very late. Actually, there are much information about the epidemiology of SARS-CoV2 . Thus the results of this work -in my opinion -are obsolete and lack of novelty.

We thank the Reviewer for the opinion, but we disagree with it. After almost 3 years of the SARS-CoV2 pandemic, we still do not have a clear view on how the restriction of international traffic affects the intensity of viral transmission in different countries and different periods of time. 

Reviewer #4: Some of the grammar can be improved. An email with comments is to be sent.

The paper is well written and understandable and the methodology has given all tools used so should be replicated if one desires to.

The paper is a bit technical in analysis but that is expected because of the analysis tools available.

A revision of the grammar should allow the paper to be published

We thank the Reviewer, and we revised the language.

In the discussion, the author should compare IBTs and OBTs introduced during the travel ban versus when the ban was lifted to be able to clearly conclude that there was no major impact..

We thank the Reviewer for this suggestion. We now conducted a binomial test to see whether the rate of cross-border transmissions were the same before and during the travel ban. The test showed that the rate of transmissions was likely to decrease during the travel ban. Nevertheless, the rate of transmissions remained noticeable. We now describe this test in the Results section.

Reviewer #5: The article: "Genomic epidemiology of SARS-CoV-2 in Russia reveals

recurring cross-border transmission throughout 2020." explains in an organized way

the effect of border closures on the transmission of SARS-COV2 infection during the first year of the pandemic.

This information is relevant because it can serve as a guiding element in decision-making by the authorities.

in health in the face of the eminent threat of a rapidly transmitted respiratory disease, as it turned out to be human coronaviruses.

We thank the Reviewer for high appreciation of the quality and importance of our research.

---

## [Decision Letter · Decision Letter 1]

28 Apr 2023

Genomic epidemiology of SARS-CoV-2 in Russia reveals recurring cross-border transmission throughout 2020

PONE-D-22-17948R1

Dear Dr. Klink,

We’re pleased to inform you that your manuscript has been judged scientifically suitable for publication and will be formally accepted for publication once it meets all outstanding technical requirements.

Kind regards,

Mohd Adnan, PhD

Academic Editor

PLOS ONE

Additional Editor Comments (optional):

Manuscript is significantly improved by the authors and now can be accepted in its current form.

Reviewers' comments:

Reviewer's Responses to Questions

**Comments to the Author**

1. If the authors have adequately addressed your comments raised in a previous round of review and you feel that this manuscript is now acceptable for publication, you may indicate that here to bypass the “Comments to the Author” section, enter your conflict of interest statement in the “Confidential to Editor” section, and submit your "Accept" recommendation.

Reviewer #1: All comments have been addressed

Reviewer #2: All comments have been addressed

2. Is the manuscript technically sound, and do the data support the conclusions?

Reviewer #1: Yes

Reviewer #2: Yes

3. Has the statistical analysis been performed appropriately and rigorously? 

Reviewer #1: N/A

Reviewer #2: Yes

4. Have the authors made all data underlying the findings in their manuscript fully available?

Reviewer #1: Yes

Reviewer #2: Yes

5. Is the manuscript presented in an intelligible fashion and written in standard English?

Reviewer #1: Yes

Reviewer #2: Yes

6. Review Comments to the Author

Reviewer #1: I am glad to see all my comments (and there were quite a few of them) have been addressed by the authors or were at least clarified in the manuscript. Images now flow better with the text as well, and authors have shown good disposition to make their manuscript more accessible in general. I think the current version is well-rounded for most readers to follow now and will prove valuable as a retrospective study for anyone looking for information on the pandemic in Russia during that period. With no further ado, I will now proceed to recommend your manuscript for publication. I wish you success.

Reviewer #2: For consistency, please double check typos. Follow MIQE guidelines for PCR data and their acronyms.

7. PLOS authors have the option to publish the peer review history of their article (what does this mean?). If published, this will include your full peer review and any attached files.

Reviewer #1: **Yes: **Rodrigo García-López

Reviewer #2: No

---

## [Editor Report · Acceptance letter]

4 May 2023

PONE-D-22-17948R1 

Genomic epidemiology of SARS-CoV-2 in Russia reveals recurring cross-border transmission throughout 2020 

Dear Dr. Bazykin:

I'm pleased to inform you that your manuscript has been deemed suitable for publication in PLOS ONE. Congratulations! Your manuscript is now with our production department. 

Kind regards, 

on behalf of

Dr. Mohd Adnan 

Academic Editor

PLOS ONE